# Systemic Scytalidium Infection with Hyperbetaglobulinemia in a Giant Schnauzer

**DOI:** 10.3390/jof11020136

**Published:** 2025-02-11

**Authors:** Andrea Grassi, Maria Elena Turba, Marianna Pantoli, Fabio Gentilini, Emanuela Olivieri, Cristian Salogni, Simona Nardoni, Matteo Gambini, Francesca Mancianti

**Affiliations:** 1Istituto Zooprofilattico Sperimentale della Lombardia e dell’Emilia-Romagna, Strada Campeggi 59-61, 27100 Pavia, Italy; emanuela.olivieri@izsler.it; 2I-Vet Diagnostica Veterinaria, Via Majorana 10, 25020 Flero, Italy; marianna.pantoli@i-vet.it (M.P.); matteo.gambini@i-vet.it (M.G.); 3Genefast srl, Via Baldassari 6, 47122 Forli, Italy; me.turba@genefast.com; 4Department of Veterinary Medical Sciences, Alma Mater Studiorum, University of Bologna, Via Tolara di Sopra 50, Ozzano dell’Emilia, 40064 Bologna, Italy; fabio.gentilini@unibo.it; 5Istituto Zooprofilattico Sperimentale della Lombardia e dell’Emilia-Romagna, Via Rovelli 53, 24125 Bergamo, Italy; cristian.salogni@izsler.it; 6Department of Veterinary Sciences, University of Pisa, Viale Delle Piagge 2, 56124 Pisa, Italy; simona.nardoni@unipi.it (S.N.); francesca.mancianti@unipi.it (F.M.)

**Keywords:** scytalidiosis, fungal infection, fungi, lymphadenopathy, systemic mycosis

## Abstract

Scytalidiosis in humans primarily causes feet and nail infections, with systemic infections rarely reported. In dogs, only one systemic infection of *Scytalidium* spp. has been reported to date. A 3-year-old giant schnauzer presented with loss of appetite, lethargy, and hind limb lameness. A complete clinical examination was performed, along with hematobiochemical tests, radiography, CT, MRI, and cytological and microbiological analyses of it enlarged lymph nodes. Hyperglobulinemia, vertebral osteolysis, and generalized lymphadenomegaly were diagnosed. Cytopathological and molecular investigations confirmed *Scytalidium*. Although treated with itraconazole, the dog’s condition worsened after a premature discontinuation of therapy, leading to euthanasia. A post-mortem and histopathological examination revealed widespread infection. This case highlights the need to consider fungal infections in cases of elevated β-2 protein.

## 1. Introduction

Fungal infections are often insidious, diagnosed late, and sometimes only detected during necropsy and they pose a significant threat to animal health due to their high mortality and morbidity rates [1]. These infections can be classified as superficial, subcutaneous, or systemic. Among these, systemic mycoses are rarely reported in animals and are mainly observed in those with a compromised immune system or underlying disorders, with only one case in the literature attributed to *Scytalidium* spp. [2]. *Scytalidium* infections typically affect the feet and nails of individuals who walk barefoot on contaminated soil in tropical regions [3]. Here, we describe a fatal case of systemic scytalidiosis in an apparently immunocompetent giant schnauzer dog, characterized by hind limb pain, spinal pain, and generalized lymphadenomegaly. Clinico-pathologically, the dog presented with severe hyperbetaglobulinemia, which was reduced following antifungal therapy.

## 2. Case Presentation

A three-year old female giant schnauzer dog was referred to the veterinary hospital due to a loss of appetite, reluctance to move, kyphosis, and left hind leg lameness. The dog resided in a rural area of the Po Valley in Northern Italy, had regular access to the outdoors and was consistently fed with commercial food.

At the time of the initial evaluation by the referring veterinarian (day 0), complete hematological and biochemical examinations were conducted. The Complete Blood Count (CBC) examination (ADVIA 2120, Siemens, Munich, Germany) revealed mild leukopenia (4.7 × 10^3^/µL; reference range 5.45–12.98 × 10^3^/µL). Reactive lymphocytes were observed in blood smears. Serum biochemical abnormalities included hyperproteinaemia (8.7 g/dL; reference range 5.2–8.5 g/dL) (AU 680, Beckman coulter, Brea, CA, USA), with an albumin/globulin ratio of 0.30. Capillary electrophoresis of serum protein (Capillarys, Sebia, Lisses, France) showed a peak in the β-2 fraction (2.48 g/dL; reference intervals 0.37–1.62 g/dL). The dog was treated with a broad-spectrum antimicrobial (cephalexin) and NSAIDs for three weeks and showed slight improvement. On day +90, due to a progressive worsening of the dog’s clinical condition, the owner returned to the practitioner.

Physical examination showed that the dog was alert and responsive with a normal respiratory rate, heart rate, and rectal temperature. At physical examination, mild lumbar discomfort was elicited by palpation and the dog showed left hind limb lameness.

Hematological and biochemical analyses revealed a further increased plasma protein concentration (10.2 g/dL; reference range 5.2–8.2 g/dL) and hypoalbuminemia (2.36 g/dL; reference range 2.6–3.80 g/dL), with an electrophoretic pattern similar to the previous findings (Figure 1a). An ELISA test against the *Leishmania infantum* antibody was negative (VetLine Leishmania, Novatec Germany, Leinfelden-Echterdingen, Germany).

### 2.1. Imaging Evaluations

Abdominal ultrasonography, computed tomography, and an MRI scan showed T3-T4 thoracic and L2-L3 lumbar discospondylitis with osteolysis and non-compressive ankylosing hyperostosis (Figure 2); generalized lymphadenomegaly with a five-fold increase in lymph node volume and a loss of structure was also observed.

### 2.2. Cytological Examination

Based on the imaging results, fine-needle aspiration was requested and carried out from the popliteal lymph node. Smears were air-dried and stained with Romanowsky stain (May–Grunwald Giemsa, Merck KGaA, Frankfurter- Darmstadt, Germany). Neutrophilic–macrophagic cell infiltrates were present, with occasionally segmented septate hyphae and, more rarely, lymphocytes, plasma cells, and eosinophilic granulocytes (Figure 3).

A diagnosis of systemic mycosis was reached and an antifungal treatment with itraconazole was initiated at 5 mg/kg orally BID. Gabapentin at 5 mg/kg PO TID was added to manage pain. After two weeks (day +105) of antifungal treatment, the owner reported that the dog was in good overall health, active, and had an improved appetite.

On day +135 (one month after the initiation of therapy), the patient’s clinical condition had significantly improved, including their total protein (7.2 g/dL) and β-2 fraction of serum proteins (1.3 g/dL) (Figure 1b). Despite the veterinarian’s recommendation to continue the therapy, the owner decided to discontinue its administration.

On day +165 (one month after the discontinuation of the therapy), the dog’s health condition deteriorated again, and she presented with neurological signs. The owner returned to the veterinary clinic, where the hematological and biochemical tests were repeated, revealing a new increase in the β-2 fraction (1.66 g/dL) and C-reactive protein (14.3 mg/dL). Following the onset of neurological clinical signs, a new treatment plan was implemented, with fluconazole at a dosage of 8 mg/kg BID. Fluconazole was chosen for its ability to cross the blood–brain barrier, although the drug is more commonly indicated for infections caused by yeasts. However, on day +175, the clinical and neurological condition of the dog had continued to worsen despite the antifungal treatment being resumed and the owner decided to proceed with euthanasia.

### 2.3. Necropsy

At necropsy, the dog was in poor physical condition, exhibiting widespread skeletal muscle hypotrophy. Gross examination revealed liver congestion with an enhanced lobular pattern, dilation of the right heart accompanied by a thinning of the right ventricular wall, and diffuse and irregular lymphadenomegaly that was most prominent in the periportal, epigastric, mesenteric, splenic, and retroperitoneal lymph nodes (Figure 4). The cross-section of the affected lymph nodes displayed irregular effacement due to miliary to coalescing white-yellowish lesions, occasionally with necrotic centres.

### 2.4. Histopathological Examination

The histopathological evaluation revealed diffuse vacuolar degeneration of the hepatocellular cords, with multifocal centrilobular necrosis in the liver; in the lymph nodes, characterized by cortical follicular and paracortical hyperplasia, severe, chronic, necrotizing, and granulomatous lymphadenitis was observed. The nodal lesions contained intralesional hyphae, ascospores, and asci within necrotic foci or within the cytoplasm of giant multinucleated histiocytes, consistent with ascomycetes (Figure 5a,b).

### 2.5. Microbiological Investigation

For the microbiological examination, a needle aspirate was obtained from the popliteal lymph nodes in vivo, and a post-mortem sample was taken from the mesenteric lymph nodes. The samples were preserved in e-Swab^®^ (Copan Italia S.p.A-Brescia, Italy) and used for fungal culture analysis. The samples were plated on Sabouraud plate agar and soy agar with 5% sheep blood (BD Diagnostic Systems, Becton Dickinson GmbH-Heidelberg, Germany). The plates were incubated at 25 °C and 37 °C. After 72 h, a pure culture of a fine, filamentous white fungal growth was observed at both incubation temperatures. By the fifth day of incubation, the culture plates were completely covered by a grey woolly growth (Figure 6a,b).

Microscopic examination of an adhesive tape preparation showed many wide and septate brown and hyaline hyphae and abundant melanized one-to-two-celled globose or barrel-shaped arthroconidia produced on the aerial mycelium (Figure 7).

The in vitro antifungal sensitivity testing was performed according to the Clinical and Laboratory Standards Institute—CLSI-M-38A2 microdilution method [4]. Antifungal activity was determined for amphotericin B, itraconazole, voriconazole and posaconazole. The minimum inhibitory concentration (MIC) for each of the four antifungal drugs tested in the present study were as follows: amphotericin B: 1 mg/L, itraconazole 4 mg/L, voriconazole 0.5 mg/L, and posaconazole 2 mg/L.

In order to characterize them from a molecular point of view, one colony was picked from the plate with a flocked swab and then DNA was extracted using a commercial kit (Maxwell RSC Blood DNA Kit, Promega Corporation, Madison, WI 53711, USA), with automated instrumentation (Maxwell RSC, Promega) following the manufacturer’s instructions and slight modifications in the form of a prolonged lysis incubation time. A negative sample, consisting of only reagents, was extracted simultaneously as a control for environmental contamination.

The DNA was then PCR-amplified using primers targeting the intergenic transcribed spacer (ITS) 1 and 2 regions [2,3]. The PCR products were then Sanger-sequenced and electrophoretically separated on an automated sequencer (ABI 310 Genetic Analyzer, Thermo Fisher Scientific, Waltham, MA 02451, USA). The sequences obtained were manually reviewed and blasted in the Genebank on the NCBI website (http://www.ncbi.nlm.nih.gov/BLAST/ (accessed on 18 May 2021)), revealing 95.46% homology with *Scytalidium lignicola*. Therefore, the isolated sample falls within the *Scytalidium* genus, but ITS region or genome sequencing is needed to improve this characterization. The negative sample did not reveal any sequences referable to *Scytalidium lignicola*.

## 3. Discussion

This report describes the clinicopathological changes, management, and outcome of a systemic *Scytalidium* spp. infection in a dog without known immunodeficiency. *Scytalidium* spp. are ascomycete fungi that are widely distributed phytopathogens primarily found in plants, fruit trees (such as lemon or banana trees), and soil [1]. The genus comprises over 15 species. *Neoscytalidium dimidiatum* and *Scytalidium hyalinum* have significant implications for human pathology [3].

*N. dimidiatum* and *S. hyalinum* can cause superficial and, less commonly, deep infections that resemble dermatophytosis, referred to as scytalidiosis. These filamentous fungi are endemic in tropical and subtropical regions like Asia, India, Africa, South America and the Caribbean, where they account for approximately 40% of dermatomycoses [5,6,7,8].

*Scytalidium* primarily affects the feet and nails of individuals walking barefoot on contaminated soils. Typically, infections remain localized to these areas; however, systemic forms characterized by central nervous system abscesses, sinusitis, osteomyelitis, mycetoma, and subcutaneous or disseminated lesions have been reported on rare occasions, especially in immunocompromised patients [3].

In cases of deep infections, *N. dimidiatum* is the most frequently implicated causative agent, while *S. hyalinum* has been found in only a few cases [9]. The prognosis of these infections is death in almost 50% of patients. There is no standardized therapy for *Scytalidium* spp. infections. Indeed, isolated fungal strains have shown high resistance to antifungal drugs commonly used in human and veterinary medicine [3]. In the present case, the fungal strain also demonstrated resistance to most antifungal agents used in human therapy.

Protein electrophoresis revealed a peak in the β-2 fraction. Proteins known to migrate to the β-2 fraction of serum include transferrin, β-2 lipoprotein, immunoglobulin M, immunoglobulin A, and positive acute-phase proteins such as C-reactive protein, complement factor 3a, and hemopexin [10]. Hypoalbuminemia can be considered a consequence of acute and/or chronic infection, considering that albumin is a negative acute-phase protein. Increased β-globulins can be observed in cases of acute or chronic inflammation [11,12]. Similar electrophoretic patterns have been observed in psittacine birds and penguins affected by aspergillosis [13,14], as well as in some case reports of infection caused by fungi other than *Aspergillus* [15].

Analogous electrophoretic alterations have been observed in some dogs affected by *Dirofilaria immitis* [16], *Mesocestoides* [17], and *Angiostrongylus vasorum* [18]. This dog had no history of travel to subtropical or endemic regions for scytalidiosis, no concomitant pathologies, and no history of immunosuppression, prolonged antibiotic use, or corticosteroid therapy. Additionally, the breed was not considered predisposed to opportunistic fungal infections [19]. The route of infection has not been identified, although the presence of more marked lymphadenomegaly in the intra-abdominal lymph nodes might suggest that the pathogen entered through the digestive tract. In other reported human cases, traumatic inoculation with plant material contaminated with the fungus has been suspected. To the best of the authors’ knowledge, this is the second report of a systemic infection caused by *Scytalidium* spp. in veterinary medicine. In the first report, which also had a fatal outcome [2], the infection affected an immunocompetent German shepherd, a dog breed known for its predisposition to systemic fungal infections [20].

## 4. Conclusions

Fungal diseases in animals are often insidious, diagnosed late, and sometimes only detected through specific diagnostic investigations. Systemic fungal infections should be considered possible differential diagnoses in dogs presenting with hyperbetaglobulinemia in their β-2 fraction. Early identification and prompt treatment of these infections are crucial to improving prognosis, as delayed diagnosis often leads to poor clinical outcomes.

## Figures and Tables

**Figure 1 jof-11-00136-f001:**
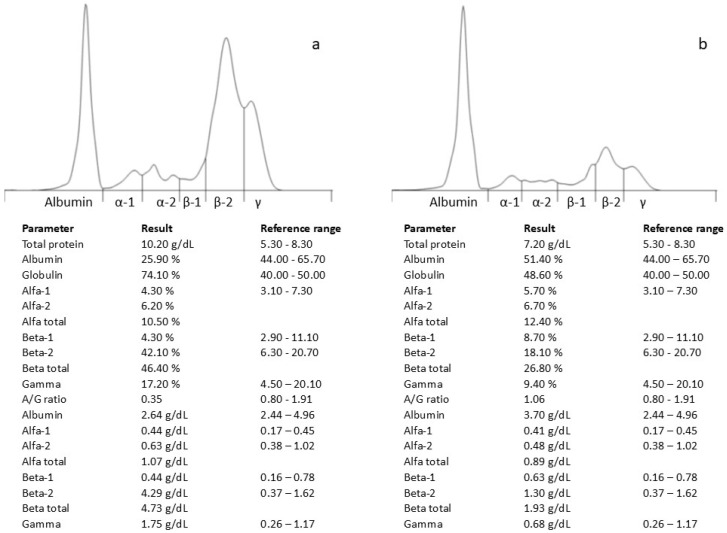
Serum protein electrophoresis profile before (**a**) and after antifungal therapy (**b**).

**Figure 2 jof-11-00136-f002:**
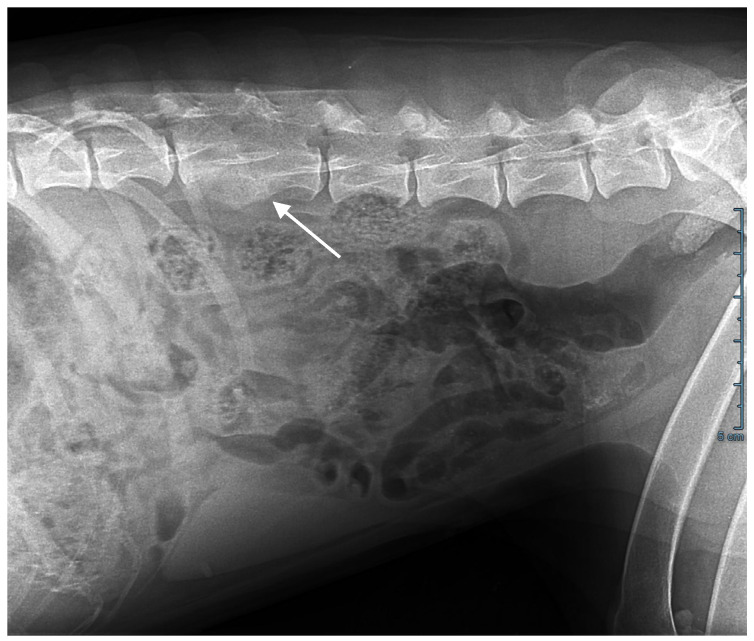
Abdominal X-ray. L2-L3 osteolysis and bone remodelling with fusion of the vertebral bodies (arrow).

**Figure 3 jof-11-00136-f003:**
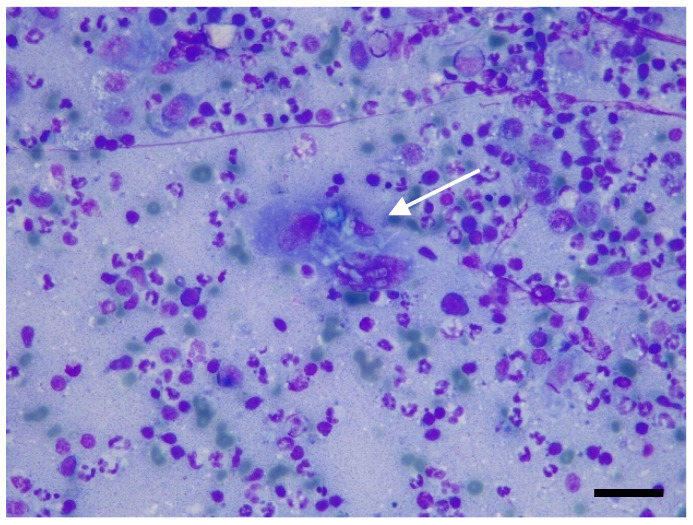
Fine-needle aspiration from popliteal lymph node: neutrophilic lymphadenitis with multinucleated giant cell containing sinuous septate fungal hyphae (arrow). May–Grünwald Giemsa stain, original magnification 200×. Scale bar: 50 μm.

**Figure 4 jof-11-00136-f004:**
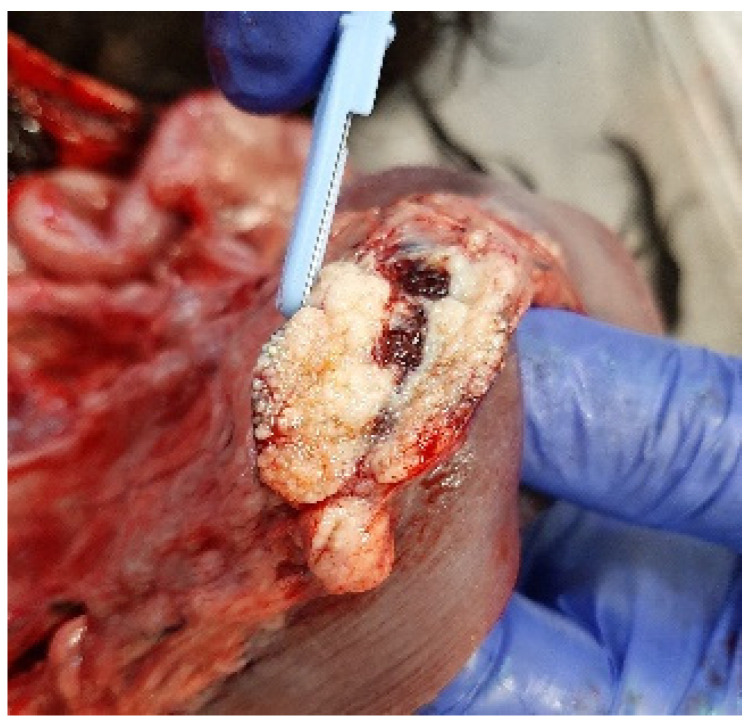
Mesenteric lymph node effaced by miliary to coalescing white-yellowish lesions.

**Figure 5 jof-11-00136-f005:**
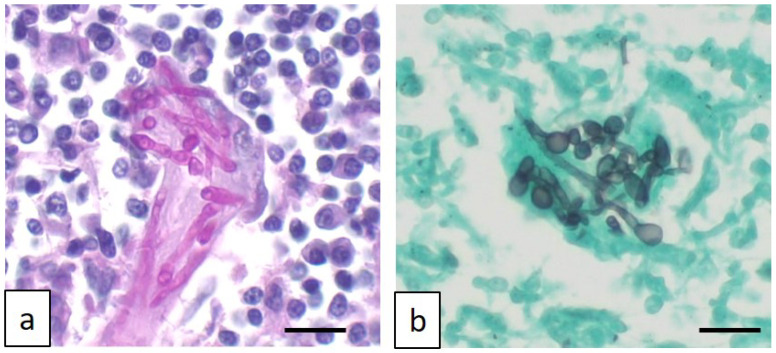
Photomicrograph of affected lymph nodes characterized by multinucleated giant histiocytes engulfing fungal sinuous hyphae, as highlighted by Periodic acid–Schiff stain (**a**) and Grocott methenamine silver stain (**b**). Original magnification 400×. Scale bar: 50 μm.

**Figure 6 jof-11-00136-f006:**
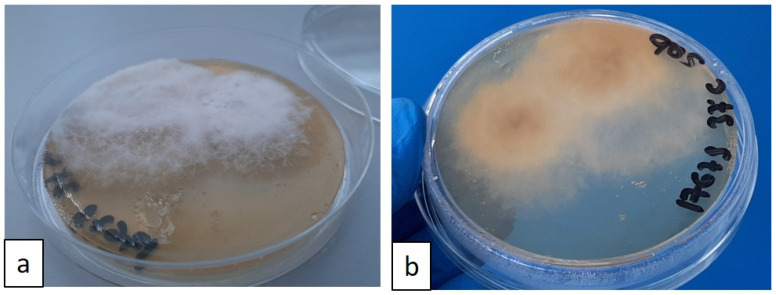
(**a**,**b**) Sabouraud agar plate, recto–verso: growth of white, cottony fungal colonies after 5 days of incubation at 37 °C; fungal colonies tend to pigment over time.

**Figure 7 jof-11-00136-f007:**
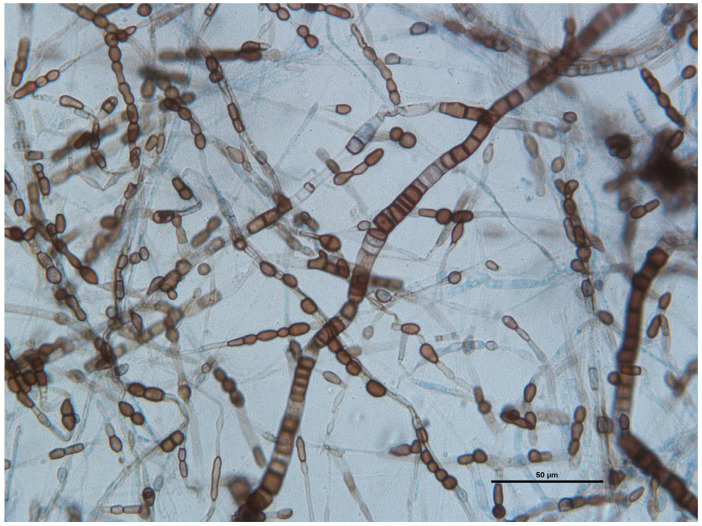
Wet-mount preparations stained with lactophenol cotton blue: septate and branched hyaline to dark brown hyphae ranging from 2 to 8 mm, with uni- or bicellular arthroconidia measuring 4 μm by 8 μm. Scale bar: 50 μm.

## Data Availability

The original contributions presented in this study are included in the article. Further inquiries can be directed to the corresponding author.

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
