# Peer review of "Systemic Scytalidium Infection with Hyperbetaglobulinemia in a Giant Schnauzer"

_jof, 2025, doi:10.3390/jof11020136_

Round 1

Reviewer 1 Report

Comments and Suggestions for Authors

Overall, this is an interesting and useful case report that touches on many of vagaries of considering and diagnosing systemic fungal diseases in dogs, especially nonendemic, opportunistic infections in otherwise clinically healthy dogs.  The amount of specific data presented on the case is very nice and detailed.  I was a little surprised there was no follow up at necropsy on the vertebral osteomyelitis, since it was an important part of the initial presentation for the case and they did two kinds of advanced imaging on it.  I did wonder why the authors chose to present a plain radiograph when both CT and MRI imaging were performed and provide enhanced visualization of osteomyelitis, osteolysis of the vertebrae, and discospodylitis.  The radiograph shown is also partially mislabeled as a thoracic radiograph when it is a radiograph of the L2-3 region.

Comments on the Quality of English Language

Overall, the quality of the English in this manuscript is very good, but there are still grammatical, punctuation and spelling errors, as well as some sentence structure problems that should be addressed for the best reading experience in English.  If the authors have assistance from a person who writes in English as a first language, they may be able to polish this easily.  This reviewer has also provided extensive notes, which might be enough since this manuscript is really quite well-written, but I don't have time to address everything.

Line 37 – grammar and punctuation errors in this line.

Line 44 - change to "hyperglobulinemia" as later in the manuscript and in the abstract and key words you refer to this more specifically as hyprbetaglobulinemia. 

Line 48 – “three,” not tree

Figure 2 -  not a a chest x-ray. Please fix figure legend.  Or consider replace with CT or MRI image of the discospondylitis.

Line 84 – typo/spelling – neutrophilic

Lines 85-86 – more errors – plasma, present

Figure 3 – scale bar is lacking; also magnification presumably 200x including the 10x ocular. This could also benefit from an arrow to point out the narrow hyphae, though this is not essential.

Line 98 – typo – spacing

Line 101 - remove capital inside the ( )

Line 102 “the”

Line 104 – 1.66 – keep the format for the decimals the same throughout the manuscript

107 – “being” not “was”

Line 124 – ascospores (pl)

Figure 5 – magnification 400x? see above

Line 138 – fix sentence – “both incubation temperatures . . .” It will be much easier to read in English.

Line 139 – either “cultures” or “a pure culture”.

Figure 7 is BEAUTIFUL!!

Line 170 – Sanger is a name in Sanger sequencing – should be capitalized; sp – “electrophoretically” 

Line 172-173 – 95% homology is not sufficient to identify this to the species, where the homology should be around 99%.  You can identify this only to the genus level with the reported  homology (See your own Reference 2). You would need more than ITS regions to identify this to the species level.  The entire sequence of S. lignicola was published in 2018 if you can do full genome sequencing to verify.  The paper does not need it unless you want to go that far with it.  ALSO – the % sign is missing on the homology number

179-180 – needs to be divided – this is 2 sentences.

Line 195 – Why dermatology? This is a systemic/deep mycosis and all the antifungals for which MICs were reported are used for systemic mycoses.  The authors pulled this concept out of context from Reference 2, the other report on Scytalidium in a dog.  Their conversation around the drugs was related to treating the dermatomycosis disease in humans whereas your discussion point is not.

Second – why was fluconazole prescribed when the dog returned with recurrent clinical signs to the veterinarian?  Fluconazole is primarily a yeast drug without much impact on hyphal fungi and usually very high MICs. Do you want to comment on that in this discussion?

Line 199- 200 – this sentence is confusing.  I am not sure what the authors are trying to tell us.

Line 205 – get rid of  “as well as an increase in total protein attributable to the beta-2 fraction reported” and put “, and” after [17]. Second, do you really want to discuss parasites here with a satisfactory amount of information on fungal infections?  Or remove parasites to their own sentence. 

208-211 – this needs splitting into at least two sentences.  It is hard to read,  Also, the breed predisposition for opportunistic fungal infections needs a reference.

Line  216-217 – 1) the reference number for the case report is incorrect according to your listed refs. 2) GSDs as a breed I believe are on the list of those predisposed to opportunistic fungal infections.  They are heavily over-represented in the literature, especially for systemic aspergillosis, but also some weird cases like Westerdykella.  If you are going to discuss genetic predisposition here, this needs a few references on what breeds are. 

Since reference 2 is mislabeled reference 5 near the end of this paper, please carefully check your numbering of the remainder of your references to be sure they are appropriately placed and numbered for manuscript.

Author Response

Overall, the quality of the English in this manuscript is very good, but there are still grammatical, punctuation and spelling errors, as well as some sentence structure problems that should be addressed for the best reading experience in English.  If the authors have assistance from a person who writes in English as a first language, they may be able to polish this easily.  This reviewer has also provided extensive notes, which might be enough since this manuscript is really quite well-written, but I don't have time to address everything.

The manuscript has been revised to improve the English

Line 37 – grammar and punctuation errors in this line.

This has been corrected

Line 44 - change to "hyperglobulinemia" as later in the manuscript and in the abstract and key words you refer to this more specifically as hyprbetaglobulinemia. 

Changed as suggested

Line 48 – “three,” not tree

This has been corrected

Figure 2 -  not a a chest x-ray. Please fix figure legend.  Or consider replace with CT or MRI image of the discospondylitis.

This has been corrected and an arrow was added. Unfortunately, we received the CT and MRI reports from the clinic, but not the images.

Line 84 – typo/spelling – neutrophilic

This has been corrected

Lines 85-86 – more errors – plasma, present

This has been corrected

Figure 3 – scale bar is lacking; also magnification presumably 200x including the 10x ocular. This could also benefit from an arrow to point out the narrow hyphae, though this is not essential.

The necessary corrections have been made

Line 98 – typo – spacing

This has been corrected

Line 101 - remove capital inside the ( )

Changed as suggested

Line 102 “the”

Changed as suggested

Line 104 – 1.66 – keep the format for the decimals the same throughout the manuscript

Changed as suggested

107 – “being” not “was”

Changed as suggested

Line 124 – ascospores (pl)

Changed as suggested

Figure 5 – magnification 400x? see above

This has been corrected

Line 138 – fix sentence – “both incubation temperatures . . .” It will be much easier to read in English.

Changed as suggested

Line 139 – either “cultures” or “a pure culture”.

Changed as suggested

Figure 7 is BEAUTIFUL!!

Thank you very much for your appreciation!

Line 170 – Sanger is a name in Sanger sequencing – should be capitalized; sp – “electrophoretically” 

Changed as suggested

Line 172-173 – 95% homology is not sufficient to identify this to the species, where the homology should be around 99%.  You can identify this only to the genus level with the reported  homology (See your own Reference 2). You would need more than ITS regions to identify this to the species level.  The entire sequence of S. lignicola was published in 2018 if you can do full genome sequencing to verify.  The paper does not need it unless you want to go that far with it.  ALSO – the % sign is missing on the homology number

The sentence was changed as follows:Therefore, the isolated sample falls within the Scytalidium genus, but for better characterization, ITS region or genome sequencing would be needed.”

179-180 – needs to be divided – this is 2 sentences.

Changed as suggested

Line 195 – Why dermatology? This is a systemic/deep mycosis and all the antifungals for which MICs were reported are used for systemic mycoses.  The authors pulled this concept out of context from Reference 2, the other report on Scytalidium in a dog.  Their conversation around the drugs was related to treating the dermatomycosis disease in humans whereas your discussion point is not.

The sentence was changed as follows: “There is no standardized therapy for Scytalidium spp. Infection. Indeed, isolated fungal strains have shown high resistance to antifungal drugs commonly used in human and veterinary medicine [3]. In the present case as well, the fungal strain demonstrated resistance to most antifungal agents used in human therapy.

Second – why was fluconazole prescribed when the dog returned with recurrent clinical signs to the veterinarian?  Fluconazole is primarily a yeast drug without much impact on hyphal fungi and usually very high MICs. Do you want to comment on that in this discussion?

Thank you for your suggestion. We have expanded on this point from lines 109-110

Line 199- 200 – this sentence is confusing.  I am not sure what the authors are trying to tell us.

We have removed this sentence

Line 205 – get rid of “as well as an increase in total protein attributable to the beta-2 fraction reported” and put “, and” after [17]. Second, do you really want to discuss parasites here with a satisfactory amount of information on fungal infections?  Or remove parasites to their own sentence. 

Thank you for your suggestion. The sentence was changed as follows: “Similar electrophoretic patterns have been observed in psittacine birds and penguins affected by aspergillosis [13,14], as well as in some case reports of infection caused by fungi other than Aspergillus [15].  Analogous electrophoretic alterations have been observed in some dogs affected by Dirofilaria immitis [16], Mesocestoides [17], and Angiostrongylus vasorum [18]”

208-211 – this needs splitting into at least two sentences.  It is hard to read,  Also, the breed predisposition for opportunistic fungal infections needs a reference.

The sentence was changed and the reference added (L 210-213)

Line  216-217 – 1) the reference number for the case report is incorrect according to your listed refs. 2) GSDs as a breed I believe are on the list of those predisposed to opportunistic fungal infections.  They are heavily over-represented in the literature, especially for systemic aspergillosis, but also some weird cases like Westerdykella.  If you are going to discuss genetic predisposition here, this needs a few references on what breeds are. 

The reference number has been corrected and these references were added

  1. HSVMA. Congenital and Heritable Disorders Guide. Aug 2012
  2. Elad D. Disseminated canine mold infections. Vet J. 2019 Jan;243:82-90. doi: 10.1016/j.tvjl.2018.11.016. Epub 2018 Nov 30

Since reference 2 is mislabeled reference 5 near the end of this paper, please carefully check your numbering of the remainder of your references to be sure they are appropriately placed and numbered for manuscript.

All the references have been checked

Reviewer 2 Report

Comments and Suggestions for Authors

This Special Issue will be devoted to studies including all the aspects of animal fungal infections, with a particular emphasis to etiology, epidemiology, pathogenesis diagnosis, control, and treatment of diseases they can induce. Experimental studies, as well as case reports and review papers, will be welcome.

The article is therefore within the scope of the special issue and is a case report.

The abstract should be expanded, especially with more mention of the material and methods.

all keywords must be different from the title.

The case report is important, but the article is written as if it were a short communication. Does JOF allow this? If accepted, the article is within this scope. However, the titles of all figures should be self-explanatory.

Author Response

The abstract should be expanded, especially with more mention of the material and methods.

We added this sentence: L24-25 A complete clinical examination was performed, along with hematobiochemical tests, radiography, CT, MRI, and cytological and microbiological analysis of the enrlarged lymph nodes. And at L28: histopathological examination.

all keywords must be different from the title.

We have change the keywords as follow: scytalidiosis, fungal infection , fungi, lymphadenopathy, systemic mycosis

The titles of all figures should be self-explanatory.

Changed as suggested

Round 2

Reviewer 1 Report

Comments and Suggestions for Authors

All recommended changes have been made to this manuscript. It is very nice.

I caught two very minor corrections:

line 139 "were" should be "was" 

line 194 after spp. "Infection" should be lower case.  

Author Response

All recommended changes have been made to this manuscript. It is very nice.

I caught two very minor corrections:

line 139 "were" should be "was" 

line 194 after spp. "Infection" should be lower case.  

Dear Reviewer,

I’ve made the corrections you suggested.

Thank you so much for your valuable feedback, it’s greatly appreciated!

Best regards,

Andrea
